# Explaining the Law of Supply and Demand via Online Learning

**Stratis Skoulakis**
Aarhus University
stratis@cs.au.dk

## Abstract

The *law of supply and demand* asserts that in a perfectly competitive market, the price of a good adjusts to a *market clearing price*. In a market clearing price $p^\star$ the number of sellers willing to sell the good at $p^\star$ equals the number of sellers willing to buy the good at price $p^\star$. In this work, we provide a mathematical foundation on the law of supply and demand through the lens of online learning. Specifically, we demonstrate that if each seller employs a no-swap regret algorithm to set their individual selling price—aiming to maximize its individual revenue—the collective pricing dynamics converge to the market-clearing price $p^\star$ . Our findings offer a novel perspective on the law of supply and demand, framing it as the emergent outcome of an adaptive learning processes among sellers.

## 1 Introduction

The *law of supply and demand* is a fundamental economic principle explaining the price of a good in a perfectly competitive market [25]. A perfectly competitive market consists of a large number of sellers and buyers where each seller is interested in selling one unit of an indistinguishable good and each buyer is interested in buying one unit of the good. The market is thus described by a supply $\mathcal{S}$ and a demand curve $\mathcal{D}$ where $\mathcal{S}(p)/\mathcal{D}(p)$ is the number of sellers/buyers willing to sell/buy the good at price $p$. The law of supply and demand states that the price of the good will converge to the *market clearing price* $p^\star$ where $\mathcal{S}(p^\star) = \mathcal{D}(p^\star)$-the number of sellers equals the number of buyers [22, 28].

The idea behind the law of supply and demand is very intuitive [1]. If the price is higher than $p^\star$ then more sellers are willing to sell than buyers willing to buy, leading to a *market surplus*. As a result, some sellers are not able to sell good and will lower their prices to attract buyers. As prices fall, more buyers are willing to buy, and fewer sellers are willing to sell. Conversely, if the price is below $p^\star$ this will create a *market shortage* that will in turn cause an upward trend of the prices. These adjustments continue until all selling prices reach $p^\star$ where there is neither market surplus or shortage [25, 22, 28].

However, upon closer examination, the explanation above does not rule out the possibility of persistent price fluctuations—where a market surplus leads to a shortage, which then causes another surplus, and so on—preventing the selling prices from ever converging to the market-clearing price. The latter raises the following fundamental question:

**Question 1.** *Why do selling prices eventually stabilize to the market clearing price $p^\star$ instead of constantly oscillating around it?*

Surprisingly, despite the *law of supply and demand* being one of the most fundamental principles in modern economics, this question remains unanswered. In this work, we provide answers to Question 1 through the lens of *game theory* and *online learning*.

---

[1] Se also https://www.investopedia.com/terms/l/law-of-supply-demand.asp?

39th Conference on Neural Information Processing Systems (NeurIPS 2025).

**Our Contribution and Techniques** As already discussed, sellers choose their prices to maximize individual revenue—that is, to sell their goods at the highest possible price. We model this price competition through a suitable pricing game (see Definition 3). In this game, each seller selects a price for their good, after which buyers arrive sequentially and purchase from the lowest-priced available seller.

The main result of this work consists in establishing that any Correlated Equilibrium (CE) [3] of the pricing game effectively coincides with the market clearing price of the market.

**Informal Theorem** *In any correlated equilibrium, the selling price is the market clearing price $p^\star$.*

To establish this result, we introduce a novel primal-dual technique. Specifically, we construct an appropriate linear program whose optimal value serves as a lower bound on the probability that a Correlated Equilibrium (CE) selects the market clearing price. We then show that this probability is at least 1 by demonstrating that the dual program admits a feasible solution with objective value 1. To the best of our knowledge, this is the first time the *dual-fitting technique* [29] is applied in the context of establishing convergence properties of learning dynamics. Thus our technique may be of independent interest.

It is well-known that in case agents use *no-swap regret algorithm* [7] to select their actions in a repeated game, then the overall joint time-average behavior converges to a Correlated Equilibrium [17, 19, 1]. We remark that assuming that sellers use no-swap regret learning is a natural assumption since these types of algorithms come with strong optimality guarantees no matter the actions of the other sellers. In view of the above the main take-away message of this work is the following:

**Take-Away Message** *If all sellers use no-swap regret algorithms to select their prices, the selling price of the good will converge to the market clearing price $p^\star$.*

As a result, our results offer a novel perspective on the law of supply and demand, framing it as the emergent outcome of a learning process among sellers. The latter reinforces the classical understanding of market equilibrium but also bridges it with contemporary decision-making algorithms.

On the negative side, we show that the weaker solution concept of Coarse Correlated Equilibrium (CCE) does not necessarily align with the market clearing price. In Theorem 3, we construct an instance of the pricing game that admits a CCE which does not coincide with the corresponding market clearing price. This implies that convergence to the market clearing price cannot be guaranteed under general no-regret dynamics, which are known to converge to CCE [20]. That being said, in Section 4, we empirically evaluate both no-regret and no-swap regret algorithms. Our experiments show that both types of learning dynamics converge to the market clearing price, suggesting that the negative result of Theorem 3 may be circumvented by specific classes of no-regret algorithms.

## 1.1 Related Work

Our work relates with the line of research studying online learning dynamics in various market settings. However none of the prior work has not studied perfectly competitive markets and the law of supply and demand.

A significant body of work has investigated *Cournot competition* under the lens of online learning. Even-Dar et al.[15] were the first to establish that no-regret dynamics converge to the respective Nash Equilibrium in the case of linear Cournot competition. Nadav et al.[26] extended these convergence results in the case of product differentiation and thet also studied *Bertrand Duopoly markets*. Fiat et al.[16] study best-response dynamics in a variant of Cournot competition where firms aim to either optimize either profit or revenue while Lin et al.[24] establish that best-response dynamics in Cournot competition converge either to a Nash equilibrium or to a periodic orbit of length two. Immorlica et al.[21] provide bounds on the Price of Anarchy in Cournot competition in case of coalitions among firms while Shit et al.[27] study the convergence properties of no-regret learning dynamics in case of limited information feedback.

Our work also relates with the line of research studying decentralized dynamics in *Fisher markets*. Bikhchandani et al. [6] showed that proportional response dynamics—an update rule closely related to online learning—converge to equilibrium in linear Fisher markets. Kolumbus et al. [23] extend these results in case of asynchronous updates. Cheung et al. [11] study the convergence properties of proportional response dynamics in Fisher markets with CES utilities while Branzei et al. [8] study

proportional dynamics in exchange economies. Finally [13, 14, 10, 9, 30] study the convergence properties of tâtonnement-type algorithms in various Fisher markets.

Babaioff et al. [4, 5] consider a pricing game very similar to ours and establish Price of Anarchy bounds on the social welfare as well as identifying structural conditions for pure Nash equilibrium in case of various types of valuations. Golrezaei [18] study the convergence properties of Online Mirror Descent (OMD) in case of firm-competition in case of consumer reference prices. Cheung et al. [12] show that in various types of markets, Online Gradient Ascent admits chaotic behavior. Zhu et al. [31] study online learning algorithms for betting markets.

## 2 Preliminaries and Results

Let $n$ denote the number of seller and buyers. We assume $k$ discrete prices $[k] = \{0, 1, \ldots, k\}$. Each seller $i \in [n]$ can sell 1 unit of good at a price no smaller than its marginal cost $s_i \in [k]$. Each buyer $j \in [n]$ is interested in buying 1 unit of good at a price at most its valuation $b_j \in [k]$. Without loss of generality we assume that $s_1 \leq \ldots \leq s_n^2$ and $b_1 \geq \ldots \geq b_n$.

The supply curve $\mathcal{S} : [k] \mapsto [n]$ denotes the number of sellers willing to sell the good at a given price, $\mathcal{S}(p) = |\{i \in [n] \ : \ s_i \leq p\}|$. The demand curve $\mathcal{D} : [k] \mapsto [n]$ denotes the number of buyers willing to buy the good at a given price, $\mathcal{D}(p) = |\{j \in [n] \ : \ b_j \geq p\}|$.

**Definition 1.** *A price $p \in [k]$ is a market clearing price if and only if $\mathcal{S}(p) = \mathcal{D}(p)$.*

We remark market clearing prices are not necessarily unique.

**Example 1.** *Consider $n = 3$ sellers and buyers. The marginal costs of the sellers are $(s_1, s_2, s_3) = (0, 2, 5)$ and the marginal prices of the buyers $(s_1, s_2, s_3) = (6, 4, 1)$. All prices $\{2, 3, 4\}$ are market clearing prices.*

Lemma 1 establishes the fact that under minimal assumption, the set of market clearing prices is not empty and in fact forms a consecutive interval.

**Lemma 1.** *In case the marginal costs and prices $s_i, b_j \in [k]$ lies in different places. Then the set of market clearing prices is not empty and is always an interval (a set of the form $\{p_1, p_1 + 1, \ldots, p_2\}$).*

*Proof.* Notice that $\mathcal{S}(k) = n$ and $\mathcal{D}(k) \leq 1$ due to the fact no two buyers' valuation can equal $k$. Similarly $\mathcal{S}(1) \leq 1$ and $\mathcal{D}(1) = n$. Thus, $\mathcal{S}(k) - \mathcal{D}(k) > 0$ and $\mathcal{S}(1) - \mathcal{D}(1) > 0$. Since marginal costs and valuations lie in different places, the difference $\mathcal{S}(p) - \mathcal{D}(p)$ can differ by at most 1 for consecutive prices. Thus, there exists $p^\star \in [k]$ such that $\mathcal{S}(p^\star) = \mathcal{D}(p^\star)$. Since $\mathcal{S}(\cdot)$ is a increasing and $\mathcal{D}(\cdot)$ is a decreasing function, the set of market clearing prices is an interval $\{p_1, p_1 + 1, \ldots, p_2\}$. $\square$

Despite the fact that strictly speaking market, clearing prices are not necessarily unique, if $k$ and $n$ are large enough and the marginal costs and prices are not very concentrated, the different market clearing prices will practical correspond to the exact same price.

**Definition 2.** *We denote with $p^\star$ the highest market clearing price.*

With some abuse of terminology in the rest of the paper we refer to $p^\star$ as the market clearing price. The law of supply and demand sates that the selling price of the good will converge to the market clearing price.

### 2.1 Pricing Games

In this section we introduce the pricing game in order to provide theoretical foundations on the law of supply and demand. Our goal is to establish that the law of supply and demand is the outcome of the strategic pricing of the sellers in their attempt to maximize their revenue.

---

[2]With a slight abuse of notation when $s_i = \ell \in [k]$, we actually mean $s_i = \ell - \epsilon$ for an arbitrarily small $\epsilon > 0$. This convention is very convenient since it ensures that seller $i \in [n]$ strictly prefer selling the good at price $s_i$ rather not selling at all.

In Definition 3 we introduce the pricing game that is an *one-shot game* capturing the competition between sellers. The strategy of each seller is the selling price that the agents selects. When selecting a price, an agent needs to balance between high revenue and the risk of not selling its good.

**Definition 3.** *In a pricing game, each seller $i$ selects a price $p_i \geq s_i$. The payoff $U_i(p_i, p_{-i})$ of seller $i \in [n]$ is defined as follows:*

- *Let $S$ denote the set of sellers, $S = \{1, \ldots, n\}$. All of them are initially available.*

- *for each buyer $j = 1$ to $n$*

  - *buyer $j$ enters the market and finds the cheapest available seller, $i_{\text{low}} := \operatorname{argmin}_{i \in S} p_i$[3].*

  - *If $p_{i_{\text{low}}} \leq b_j$, then*

    1. *Buyer $j$ buys the good from $i_{\text{low}}$ at price $p_{i_{\text{low}}}$.*
    2. *Seller $i_{\text{low}}$ gets utility $U_{i_{\text{low}}}(p_i, p_{-i}) := p_{i_{\text{low}}} - s_{i_{\text{low}}}$ and exits $S \leftarrow S/\{i_{\text{low}}\}$.*

*A seller $i \in [n]$, that did not sell its good, gets utility $U_i(p_i, p_{-i}) = 0$.*

We remark that in the pricing game of Definition 3 only sellers are strategic agents. When a buyer enters the market, it considers the lowest-priced seller who is still available at that moment. We emphasize that each seller can supply only 1 unit of the good. That is why once a seller sells its unit, then it exits the market.

We denote with $\mathcal{P}_i$ the strategy space of seller $i \in [n]$, $\mathcal{P}_i = \{s_i, \ldots, k\}$. We also denote with $\mathcal{P} := \mathcal{P}_1 \times \mathcal{P}_2 \times \cdots \times \mathcal{P}_n$ the set of pricing profiles. For a pricing profile $p := (p_1, \ldots, p_n) \in \mathcal{P}$ we also use the notation $p = (p_i, p_{-i})$ to denote the price of seller $i \in [n]$ with the prices selected by the other sellers.

## 2.2 No-Swap Regret Minimization and Correlated Equilibrium

To model the market's behavior over time, we consider sellers repeatedly playing the pricing game of Definition 3 across multiple rounds. We remark that the beginning of each round all sellers admit 1 unit of good regardless of whether they performed a sale in the previous round.

---

**Protocol 1:** Pricing Game over time

At each round $t = 1, \ldots, T$
- Each seller $i \in [n]$, (secretly) selects a price $p_i^t \in \mathcal{S}_i$.
- Each seller $i \in [n]$, gets utility $U_i(p_i^t, p_{-i}^t)$ (see Definition 3).
- Each seller $i \in [n]$, learns $p_{-i}^t$ and uses this information to select its next price $p_i^{t+1}$.

---

A seller $i \in [n]$, needs to come up with a *pricing strategy* that at each round $t \in [T]$ selects a price $p_i^t$ solely based on past prices $p_{-i}^1, \ldots, p_{-i}^{t-1} \in [k]$. The online learning framework provides such decision-making algorithms that base their decision on prior observations [20].

In a pricing games an online learning algorithm $\mathcal{A}$, at round $t \in [T]$, produces a mixed strategy $x_i^t \in \Delta(\mathcal{P}_i)$ over a set of possible prices $\mathcal{P}_i$. The performance of an online learning algorithm $\mathcal{A}$ can be quantified through the notion of *regret* [20, 7].

**Definition 4.** *The regret of an online learning algorithm $\mathcal{A}$ is defined as*

$$\mathcal{R}_\mathcal{A}(T) \quad := \quad \max_{p_i^\star \in \mathcal{P}_i} \sum_{t=1}^{T} \mathbb{E}_{p_i \sim x_i^t} \left[ U_i(p_i^\star, p_{-i}^t) \right] - \sum_{t=1}^{T} \mathbb{E}_{p_i \sim x_i^t} \left[ U_i(p_i, p_{-i}^t) \right]$$

---

[3]In case of tie, buyer $j \in [n]$ selects the seller with the highest index.

*The swap regret of an online learning algorithm $\mathcal{A}$ is defined as*

$$\mathcal{R}_{\mathcal{A}}^{swap}(T) \quad := \quad \max_{\delta:\mathcal{P}_i \mapsto \mathcal{P}_i} \sum_{t=1}^{T} \mathbb{E}_{p_i \sim x_i^t}\left[ U_i(\delta(p_i), p_{-i}^t) \right] - \sum_{t=1}^{T} \mathbb{E}_{p_i \sim x_i^t}\left[ U_i(p_i, p_{-i}^t) \right]$$

*Algorithm $\mathcal{A}$ is called no-regret if $\mathcal{R}_{\mathcal{A}}(T) = o(T)$ no matter the prices $p_{-i}^1, \ldots, p_{-i}^T$ selected by the other sellers. Respectively if $\mathcal{R}_{\mathcal{A}}^{\mathrm{swap}}(T) = o(T)$ the algorithms is called no-swap regret.*

A no-regret algorithm $\mathcal{A}$ guarantees that its time-averaged utility converges to the the time-averaged utility produced by the *best fixed price $p_i^\star \in \mathcal{P}_i$* regardless the prices selected by the other providers. A no-swap regret algorithm provides the stronger optimality guarantees that the time-average payoff is at least the time-average payoff of the *best fixed switching function $\delta(\cdot)$*.

It is well known that if all sellers use no-regret algorithms to select their prices, the overall behavior converges to a *Coarse Correlated Equilibrium*[2]. Similarly, if sellers use no-swap regret algorithms, the resulting dynamics converge to a *Correlated Equilibrium* [19, 17]. Both equilibrium concepts are formally defined in Definition5.

**Definition 5.** *A probability distribution $\mu \in \Delta(S)$ over pricing profiles $\mathcal{P} := \times_{i \in [n]} \mathcal{P}_i$, is a Coarse Correlated Equilibrium if for each seller $i \in [n]$,*

$$\mathbb{E}_{p \sim \mu}\left[ U_i(p_i, p_{-i}) \right] \geq \max_{p_i' \in \mathcal{P}_i} \mathbb{E}_{p \sim \mu}\left[ U_i(p_i', s_{-i}) \right].$$

*A probability distribution $\mu \in \Delta(S)$ over pricing profiles $\mathcal{P} := \times_{i \in [n]} \mathcal{P}_i$, is a Correlated Equilibrium if for each seller $i \in [n]$,*

$$\mathbb{E}_{p \sim \mu}\left[ U_i(p_i, p_{-i}) \right] \geq \max_{\delta:\mathcal{P}_i \mapsto \mathcal{P}_i} \mathbb{E}_{p \sim \mu}\left[ U_i(\delta(p_i), p_{-i}) \right].$$

## 2.3 Our Results and Paper Organization

To this end, we present the main contribution of our work: establishing that the law of supply and demand can be viewed as the limiting behavior of no-swap regret algorithms employed by sellers to maximize their individual payoffs.

**Theorem 1.** *Let a pricing game with market clearing price $p^\star$ and $\mu \in \Delta(\mathcal{P})$ be a Correlated Equilibrium. If $(p_1, \ldots, p_n) \sim \mu$ then with probability 1,*

1. *all sellers $i \in [n]$ with $s_i \leq p^\star$, select $p_i = p^\star$ and sell their good.*

2. *all buyers $j \in [n]$ with $b_j \geq p^\star$, buy the good at price $p^\star$.*

3. *all sellers $i \in [n]$ with $s_i > p^\star$ do not sell anything and all buyers $j \in [n]$ with $b_j < p^\star$ do not buy anything.*

Theorem 1 establishes that the selling price of the good is the market clearing price $p^\star \in [k]$. This is because executed sales occur at price $p^\star$. Notice that Theorem 1 establishes that sellers $i \in [n]$ with marginal costs higher than $s_i > p^\star$ do not sell their good. Similarly, buyers $j \in [n]$ with marginal prices $b_j < p^\star$ never buy the good since $p^\star$ is the lowest price offered in the market.

Combining Theorem 1 with the known convergence results of no-swap regret dynamics [17, 19] we get Corollary 1, showing that the law of supply and demand emerges naturally when sellers use no-swap regret algorithms. This is the main takeaway of our work.

**Corollary 1.** *If all sellers in a pricing game use a no-swap regret to select their prices, then the price of the good converges to the market-clearing price $p^\star \in [k]$.*

In Theorem 2 we show that Coarse Correlated Equilibria (CCE) is not always compatible with the market clearing price. In particular we show that there are instances of the pricing game admitting CCEs at which the probably of sellers selecting the market clearing price is arbitrarily close to zero.

**Theorem 2.** *There exists a family of pricing games with $n = 2$ admitting a Coarse Correlated Equilibrium $\mu \in \Delta(\mathcal{P})$ such that the probability that any of the sellers plays a market clearing price is at most $\mathcal{O}(1/k)$ where $[k]$ is the set of prices.*

The proof of Theorem 2 is deferred to Appendix A. In Section 3 we present the highlevel ideas behind the proof of Theorem 1. Finally in Section 4 we experimentally evaluate the convergence properties of both no-regret and no-swap regret algorithm in the context of pricing games.

# 3 Proof of Theorem 1

In this section, we present the proof of Theorem 1, which is based on a *dual-fitting* argument [29]. To the best of our knowledge, this is the first application of such techniques to establish convergence results, and it may be of independent interest.

We first introduce some necessary definitions. We remind that $p^\star$ denotes the maximum market clearing price, see Definition 2.

**Definition 6.** *We call a pricing profile $p = (p_1, \ldots, p_n) \in \mathcal{P}$ valid if and only if $p_i = p^\star$ for all sellers $i \in [n]$ with $s_i \leq p^\star$. We also denote with $\mathcal{V}$ the set of all valid pricing profiles.*

Using Definition 6 we can restated Theorem 1 as of the fact that any Correlated Equilibrium $\mu \in \Delta(\mathcal{S})$, places all of its probability mass on *valid* pricing profiles. The latter is formally stated in Theorem 3 and consists the main technical contribution of our work.

**Theorem 3.** *Let $\mu \in \Delta(\mathcal{S})$ be a Correlated Equilibrium of a pricing game. Then, $\mu(p) > 0$ if and only if $p \in \mathcal{V}$.*

In the rest of the section we present the proof of Theorem 3 through our dual fitting technique.

**Definition 7.** *Let the function $r : \mathcal{P} \mapsto \{0, 1\}$ on the pricing profiles: $r(p) = 1$ for all $p \in \mathcal{V}$ and $r(p) = 0$ for any $p \notin \mathcal{V}$.*

Using the reward function defined above, we introduce the following linear program that we will play a key role in our proof.

**Definition 8.** *Given a pricing game of Definition 3, consider the following linear program,*

$$\min \quad \sum_{p \in \mathcal{P}} r(p) \cdot \mu(p)$$

$$s.t. \quad \sum_{p_{-i} \in \mathcal{P}_{-i}} \mu(p_i, p_{-i}) \left( U_i(p'_i, p_{-i}) - U_i(p_i, p_{-i}) \right) \leq 0 \quad \text{for all } i \in [n] \text{ and } p_i, p'_i \in \mathcal{P}_i$$

$$\sum_{p \in \mathcal{P}} \mu(s) = 1$$

$$\mu(p) \geq 0 \quad \forall p \in \mathcal{P}$$

*where $U_i(p_i, p_{-i})$ is the utility of seller $i \in [n]$. We also denote with $Z^\star_{LP}$ its optimal value.*

The idea behind the linear program of Definition 8 is that any Correlated Equilibrium $\mu \in \Delta(\mathcal{P})$ will satisfies its constraints. One can show that $\mu \in \Delta(\mathcal{P})$ satisfies the first constraint by using the fact that $\mathbb{E}_{p \sim \mu} [U_i(p_i, p_{-i})] \geq \mathbb{E}_{p \sim \mu} [U_i(\delta(p_i), p_{-i})]$ for the switching function with $\delta(s_i) = s'_i$. The rest of the constraints are satisfied due to the fact that $\mu(\cdot)$ is a joint probability distribution. The latter is formally stated and established in Lemma 6.

**Lemma 2.** *Let $\mu \in \Delta(\mathcal{P})$ be a Correlated Equilibrium. Then $\mu \in \Delta(\mathcal{P})$ satisfies the constraints of the linear program of Definition 8 and thus $\sum_{p \in \mathcal{P}} \mu(p) \cdot r(p) \geq Z^\star_{LP}$.*

Our cornerstone idea is that the optimal value of the linear program above acts as a lower bound on the probability of sampling a valid pricing profile. The latter is formally stated in Corollary 2.

**Corollary 2.** *Let $\mu \in \Delta(\mathcal{P})$ be a Correlated Equilibrium. Then $\Pr_{p \sim \mu}[\, p \text{ is valid} \,] \geq Z^\star_{LP}$.*

*Proof.* By Definition 7, $r(p) = 1$ if $p$ is valid and 0 otherwise. Thus, $\sum_{p \in \mathcal{P}} \mu(p) \cdot r(p) = \sum_{p \in \mathcal{V}} \mu(p) \cdot r(p) = \Pr[p \text{ is valid}]$. $\square$

To complete our proof we just need to establish that $Z^\star_{LP} \geq 1$. We will show that the optimal value of the dual of the linear program of Definition 7 is at least 1. Then the claim follows by weak duality.

**Lemma 3.** *The following LP is the dual of the program in Definition 8. We denote with $D^\star$ its optimal value.*

$$\max \quad \mu$$

$$s.t. \quad \mu + \sum_{i \in [n]} \sum_{p'_i \in \mathcal{P}_i} \lambda^i_{p_i p'_i} \left( U_i(p_i, p_{-i}) - U_i(p'_i, p_{-i}) \right) \leq r(p) \quad \text{for all } p \in \mathcal{P}$$

$$\lambda^i_{p_i p_{-i}} \geq 0 \quad \forall i \in [n], p_i, p_{-i} \in \mathcal{P}_i$$

**Lemma 4.** *The optimal value of the dual linear program is at least 1, $D^\star \geq 1$.*

Lemma 4 is the main technical contribution of this section. In Section 3.1, we present a sketch of its proof. in Section 3.1. We complete this section with the proof of Theorem 3.

*Proof of Theorem 3.* By Corollary 2 we know that $\Pr_{p \sim \mu}[\, p \text{ is valid}\,] \geq Z^\star_{LP} \geq D^\star \geq 1$ where the fact that $Z^\star_{LP} \geq D^\star$ is due to weak duality. $\qquad\square$

## 3.1 The Dual-Fitting Argument

In this section we present the proof of Lemma 4 stating that the optimal value of the dual program is at least 1, $D^\star \geq 1$.

Our approach is to select an assignment of the dual variables $\{\mu, \lambda^i_{s_i, s_{-i}}\}$ that are dual feasible and at the same time $\mu = 1$. Our assignment is presented in Definition 9 that we denote with $\hat{\lambda}^i_{s_i s'_i}$ in order to differentiate it from the variables.

**Definition 9.** *For any seller $i \in [n]$ with $s_i \leq p^\star$,*

$$\hat{\lambda}^i_{p_i p'_i} := \begin{cases} 0 & \text{if } p'_i \neq p^\star \\ (2nk)^{2np_i - i} & \text{if } p'_i = p^\star \end{cases}$$

*For any seller $i \in [n]$ with $s_i > p^\star$, $\hat{\lambda}^i_{p_i p'_i} = 0$.*

Before proceeding, let us provide the high-level intuition behind the assignment of Definition 9. The dual variable $\lambda^i_{p_i p'_i}$ represents a deviation of seller $i \in [n]$ from price $p_i$ to price $p'_i$. By setting $\hat{\lambda}^i_{p_i p'_i} > 0$ only when $p_i = p^\star$ and 0 otherwise, we enforce that all agents $i \in [n]$ with $p_i \leq p^\star$ are incentivized to deviate exclusively to price $p^\star$, and not to any other price.

Next, we establish that by selecting the assignment $\hat{\lambda}^i_{s_i s'_i}$ as of Definition 9, we can select $\hat{\mu} = 1$ and the overall assignment $(1, \hat{\lambda})$ is dual feasible. To establish the latter, for any pricing profile $p \in \mathcal{P}$ we consider

$$\hat{b}_p := r(p) - \sum_{i \in [n]} \sum_{p'_i \in \mathcal{S}_i} \hat{\lambda}^i_{p_i p'_i} \cdot (U_i(p_i, p_{-i}) - U_i(p'_i, p_{-i})) \,.$$

We will establish that $\hat{b}_p \geq 1$ for all pricing profiles $p \in \mathcal{P}$. Then it directly follows that $(1, \hat{\lambda})$ is feasible for the dual, and thus $D^\star \geq 1$, since the dual is a maximization linear program. The fact that $\hat{b}_p \geq 1$ for all $p \in \mathcal{P}$ is respectively established in Lemma 5 and Lemma 6.

**Lemma 5.** *Let a valid pricing profile $p \in \mathcal{V}$. Then $b_p = 1$.*

*Proof.* Since $p \in \mathcal{P}$ is a valid pricing profile, $r(p) = 1$. As a result, we only need to establish that $\sum_{i \in [n]} \sum_{p'_i \in \mathcal{S}_i} \hat{\lambda}^i_{p_i p'_i} \cdot (U_i(p_i, p_{-i}) - U_i(p'_i, p_{-i})) = 0$. This follows by the fact that $\hat{\lambda}^i_{p_i p'_i} = 0$ for all sellers $i \in [n]$ with $s_i > p^\star$ and for all sellers $i \in [n]$ with $s_i \leq p^\star$, $p_i = p^\star$ meaning that

$$\sum_{p'_i \in \mathcal{S}_i} \hat{\lambda}^i_{p_i p'_i} \cdot (U_i(p_i, p_{-i}) - U_i(p'_i, p_{-i})) = \hat{\lambda}^i_{p_i p^\star} \cdot (U_i(p_i, p_{-i}) - U_i(p^\star, p_{-i})) = 0.$$

$\qquad\square$

We complete the section with Lemma 6-establishing the respective claim for non-valid pricing profiles $p \notin \mathcal{V}$. The full proof of Lemma 6 is presented to Appendix C.

**Lemma 6.** *Let a non valid pricing profile $p \notin \mathcal{V}$. Then $b_p \geq 1$.*

*Sketch of Proof.* Since the pricing profile $p \notin \mathcal{V}$ is not valid, by Definition 7 we get that $r(p) = 0$. As a result, we need to establish that

$$\sum_{i \in [n]} \sum_{p'_i \in \mathcal{S}_i} \hat{\lambda}^i_{p_i p'_i} \cdot (U_i(p'_i, p_{-i}) - U_i(p_i, p_{-i})) \geq 1.$$

Since $p^\star \in [n]$ is the highest market clearing price, $p^\star + 1$ is not a market clearing price. Thus, $p^\star = b_j$ for some buyer $j \in [n]$ or $p^\star + 1 = s_i$ for some seller $i \in [n]$.

Let us here consider the case $p^\star = b_j$. To simplify notation let $m := |\{i \in [n] \ : \ s_i \leq p^\star\}|$ and $i^\star$ the highest price, $i^\star := \operatorname{argmax}_{\{i \in [n] \ : \ s_i \leq p^\star\}} p_i$.

Notice that in case $p_{i^\star} \geq p^\star + 1$ then seller $i^\star$ does not set its good and thus $U_{i^\star}(p_{i^\star}, p_{-i^\star}) = 0$. This is because at $m - 1$ buyers are willing to buy the good at price $p_{i^\star} \geq p^\star + 1$ (recall that $p^\star = b_j$) and they are at least $m - 1$ sellers with lower prices. At the same time, seller $i^\star$ always sells its good in case $p_{i^\star} = p^\star$. This is because there are $m$ buyer willing to buy the good at price $p^\star$. Thus, $U_{i^\star}(p^\star, p_{-i^\star}) = p^\star - s_{i^\star} \geq 1$. As a result,

$$\hat\lambda^i_{p_{i^\star} p^\star} \cdot (U_{i^\star}(p^\star, p_{-i^\star}) - U^\star_i(p_{i^\star}, p_{-i^\star})) \geq (2nk)^{2np_{i^\star} - i^\star}. \tag{1}$$

Since $i^\star := \operatorname{argmax}_{\{i \in [n] \ : \ s_i \leq p^\star\}} p_i$ we know that $p_i \geq p_{i^\star}$ for all sellers $i \in [n]$ with $s_i \leq p^\star$. To simplify things here let us assume that $p_i > p_{i^\star}$ (the case of ties introduces some additional complications and is presented in the full proof).

Let us now try to identify the worst-case for the quantity $U_i(p^\star, p_{-i}) - U_i(p_i, p_{-i})$. First notice that $U_i(p_i, p_{-i}) \leq p_i - s_i$ since otherwise $U_i(p_i, p_{-i}) = 0$ and that $U_i(p^\star, p_{-i}) = p^\star - s_i$. Thus,

$$\sum_{i \neq i^\star} \hat\lambda^i_{p_i p^\star} \cdot \left( \underbrace{U_i(p^\star, p_{-i})}_{p^\star - s_i} - \underbrace{U_i(p_i, p_{-i})}_{\leq p_i - s_i} \right) \geq \sum_{i \neq i^\star} (2nk)^{2np_i - i} \cdot (p^\star - p_i)$$

$$\geq -n(2nk)^{2n(p_{i^\star} - 1) - 1} k \geq -(2nk)^{2n(p_{i^\star} - 1)} \tag{2}$$

where the last inequality comes from the fact that $p_i \leq p_{i^\star} - 1$ and that $p^\star - p_i \leq k$. By Equations 1 and 2 we get that

$$\sum_{i \in [n]} \sum_{p'_i \in \mathcal{S}_i} \hat\lambda^i_{p_i p'_i} \cdot (U_i(p'_i, p_{-i}) - U_i(p_i, p_{-i})) \geq (2nk)^{2np_{i^\star}} \left( (2nk)^{-i^\star} - (2nk)^{-2n} \right) \geq 1.$$

$\square$

# 4 Experimental Evaluations

In this section we experimentally evaluate the well-known Hedge [20] no-regret algorithm and the no-swap regret algorithm proposed by Blum and Mansour [7]. We consider as the set of prices the $[0, 5]$ interval with 0.2 discretization-there are the following 30 possible prices $\{0, 0.2, \dots, 4.8, 5\}$.

We first consider the family of instances of the pricing game constructed to establish Theorem 2 (see also Appendix A). This instance is composed by $n = 2$ sellers and buyers where $(s_1, s_2) = (0, 0)$ and $(b_1, b_2) = (5, \lambda)$. As a result, the highest market clearing price is $p^\star = \lambda$. Despite the fact that such instances admit CCEs that do not correspond to any market clearing price, our experimental evaluation reveal that the Hedge algorithm always converges to the market clearing price $p^\star$, see Figure 1,2 and 3.

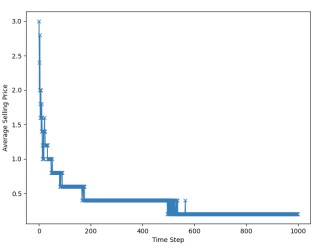
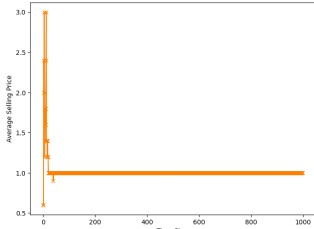
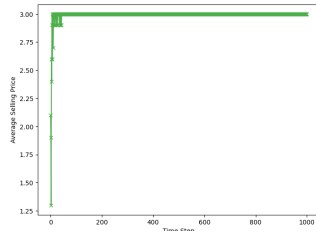

Figure 1: $\lambda = 0.2$      Figure 2: $\lambda = 1$      Figure 3: $\lambda = 3$

Next we consider a more natural set-up with $n = 100$ sellers and sellers. We consider the linear demand curve $\mathrm{D}(p) = -20p + 100$ for $p \in [0, 5]$ and three different supply curves, $\mathcal{S}_{\mathrm{linear}}(p) :=$

$20p, \mathcal{S}_{\text{quad}}(p) := p^2/0.25$ and $\mathcal{S}_{\text{linear}}(p) := p/5, \mathcal{S}_{\text{sqrt}}(p) := 100\sqrt{p/5}$ (see Figure 4). Each supply curve intersects with the demand curve at a different price, resulting in different market clearing prices. As Figure 5 depicts, if the sellers use the Hedge algorithm, the average selling price converges fast to the respective market clearing price.

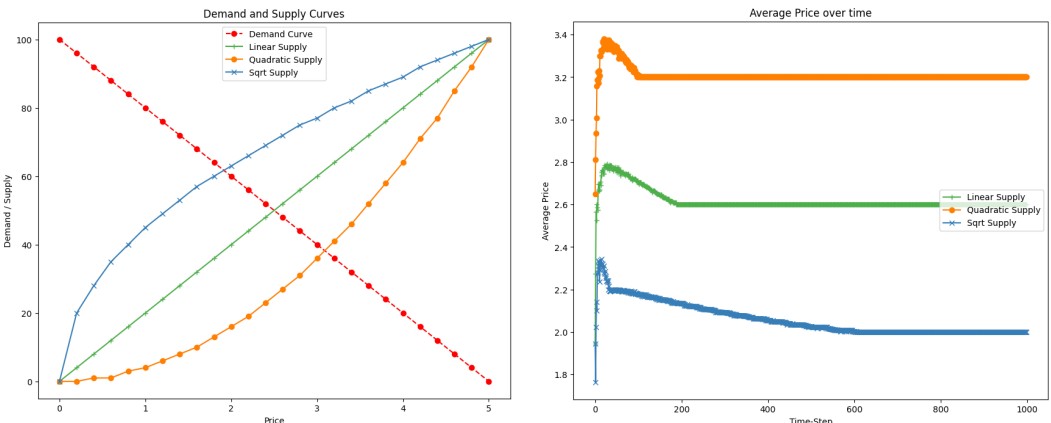

Figure 4: Supply and Demand Curves        Figure 5: Average Selling Price

The performance of the no-swap regret algorithm of Blum et al. [7] is very similar with the performance of Hedge. Since convergence to the market clearing price is ensured by Theorem 1, the respective figures are deffered to Appendix D. In Appendix D we also present additional experimental evaluations for other supply/demand curves.

## 5 Conclusion

In this paper, we establish a rigorous connection between the law of supply and demand and the dynamics of online learning within adequate pricing games. Our main contribution is establishing that any correlated equilibrium (CE) of the associated pricing game aligns with the market-clearing price. The latter implies that when all sellers employ no-swap regret algorithms, the price converges to the market clearing price. Thus, our results provide an interesting theoretical foundation on the law of supply and demand by framing it as the emergent behavior of a learning process among the sellers.

While we establish that coarse correlated equilibria (CCE) do not inherently guarantee convergence to the market-clearing price, our experimental evaluations indicate that once sellers use the well-known Hedge algorithm, the resulting price dynamics converge to the market clearing price. As a result, it is likely that a certain classes of no-regret algorithms, such as mean-based algorithms, may be able to always converge to the market clearing price. Providing formal theoretical convergence guarantees for specific classes of no-regret algorithms, is a very interesting research direction.

**Limitations** Establishing that general no-regret dynamics converge to the market-clearing price would offer a stronger theoretical foundation for the law of supply and demand. However, as Theorem 2 demonstrates, Coarse Correlated Equilibria do not always align with the market-clearing price, so such convergence properties cannot hold for all no-regret sequences of play. However our experiments suggest that the Hedge algorithm consistently reaches the market-clearing price. Extending our convergence results to specific clasees of no-regret dynamics remains an open challenge and a limitation of this work.

**Broader Impact** We acknowledge that there are many potential societal consequences of our theoretical results, however none of which we feel must be specifically highlighted.

**Acknowledgments** This project was supported by the Villum Young Investigator Award (Grant no. 72091).

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

## A  Proof of Theorem 2

**Theorem 2.** *There exists a family of pricing games with $n = 2$ admitting a Coarse Correlated Equilibrium $\mu \in \Delta(\mathcal{P})$ such that the probability that any of the sellers plays a market clearing price is at most $\mathcal{O}(1/k)$ where $[k]$ is the set of prices.*

*Proof.* Consider a pricing game of Definition 3 with $n = 2$ such that $s_1 = s_2 = 0$ and $b_1 = 1$ and $b_2 = k$. To this end notice that the market clearing prices of the game are either $0$ or $1$. We will show that there exists a Coarse Correlated Equilibrium $\mu \in \Delta(\mathcal{P})$ such that

$$\Pr_\mu[s_1 \in \{0, 1\}] + \Pr_\mu[s_2 \in \{0, 1\}] \le \mathcal{O}\left(\frac{1}{k}\right).$$

To simplify notation let $k := 2\ell + 1$. Up next we define the joint probability distribution $\mu \in \Delta(\mathcal{P})$ as follows:

1. randomly select $s \sim \mathrm{Unif}(1, \dots, k)$.

2. then $\Pr[(s, (s-1) \bmod k)] = \frac{\ell+1}{2\ell+1}$ and $\Pr[((s-1) \bmod k), s)] = \frac{\ell}{2\ell+1}$.

Notice that due to the symmetry $\mu(\cdot)$, $\Pr_\mu[s_1 < s_2] = (\ell+1)/(2\ell+1)$ and $\Pr_\mu[s_2 > s_1] = \ell/(2\ell+1)$. The latter implies that for seller 1 that,

$$\mathrm{E}_\mu[U_1(s_1, s_2)] = \frac{\ell+1}{2\ell+1} \frac{\sum_{i=1}^{2\ell+1} i}{2\ell+1} = \frac{(\ell+1)^2}{2\ell+1}$$

Similarly for seller 2 we get that

$$\mathrm{E}_\mu[U_2(s_1, s_2)] = \frac{\ell}{2\ell+1} \frac{\sum_{i=1}^{2\ell+1} i}{2\ell+1} = \frac{\ell(\ell+1)}{2\ell+1}$$

Now let assume that seller 1 deviates to a fixed price $i$. In this case

$$\mathrm{E}_\mu[U_1(i, s_2)] = i \cdot \Pr_\mu[s_2 \ge i] = i \cdot \left(1 - \frac{i}{2\ell+1}\right) = i \cdot \frac{2\ell+1-i}{2\ell+1} \le \frac{(\ell+1)^2}{2\ell+1} = \mathrm{E}_\mu[U_1(s_1, s_2)]$$

Now let assume that seller 2 deviates to fixed price $i$. In this case

$$\mathrm{E}_\mu[U_2(i, s_1)] = i \cdot \Pr_\mu[s_1 > i] = i \cdot \left(1 - \frac{i-1}{2\ell+1}\right) = i \cdot \frac{2\ell+2-i}{2\ell+1} \le \frac{\ell(\ell+2)}{2\ell+1} = \mathrm{E}_\mu[U_2(s_2, s_1)]$$

As a result, the joint probability distribution $\mu \in \mathcal{P}$ is a Coarse Correlated Equilibrium (CCE) while probability that the price $0$ or $1$ (the set of market clearing prices) is playd by either seller 1 or 2 is at most $\mathcal{O}(1/k)$.

$\square$

## B  Omitted Proofs of Section 3

**Lemma 2.** *Let $\mu \in \Delta(\mathcal{P})$ be a Correlated Equilibrium. Then $\mu \in \Delta(\mathcal{P})$ satisfies the constraints of the linear program of Definition 8 and thus $\sum_{p \in \mathcal{P}} \mu(p) \cdot r(p) \ge Z_{LP}^\star$.*

*Proof.* Since $\mu \in \Delta(\mathcal{P})$ then $\sum_{p \in \mathcal{P}} \mu(p) = 1$ and $\mu(p) \ge 0$ for all $p \in \mathcal{P}$. Notice that by Definition 5 we know that for any seller $i \in [n]$ and any switching function $\delta : \mathcal{P}_i \mapsto \mathcal{P}_i$,

$$\sum_{p_i} \sum_{p_{-i}} \mu(p_i, p_{-i}) \cdot (U_i(p_i, p_{-i}) - U_i(\delta(p_i), p_{-i})) \ge 0$$

Let $p_i, p_i' \in \mathcal{P}_i$ and consider the switching function $\delta(p_i) = p_i'$ and $\delta(x) = x$ otherwise. Then we directly get that

$$\sum_{p_{-i}} \mu(p_i, p_{-i}) \cdot (U_i(p_i, p_{-i}) - U_i(p_i', p_{-i})) \ge 0 \quad \text{for all } p_i, p_i' \in \mathcal{P}.$$

$\square$

**Lemma 4.** *The optimal value of the dual linear program is at least* $1$, $D^\star \geq 1$.

*Proof.* By taking the Lagragian

$$L := \sum_p \mu(p) \cdot r(p) - \sum_{i,p_i,p'_i} \lambda^i_{p_i,p'_i} \left( \epsilon + \sum_{p_{-i}} \mu(p_i, p_{-i}) \cdot (U_i(p_i, p_{-i}) - U_i(p'_i, p_{-i})) \right) + \mu \left( 1 - \sum_p \mu(p) \right) - \sum_p k_p \cdot \mu(p)$$

where $\lambda^i_{p_i,p_{-i}}, k_p \geq 0$. By rearranging the terms we get that

$$L := \sum_p \mu(p) \left( r(p) + \sum_{p'_i} \lambda^i_{p_i,p'_i}(U_i(p'_i, p_{-i}) - U_i(p_i, p_{-i})) - \mu - k_p \right) + \mu - \epsilon \sum_{i,p_i,p'_i} \lambda^i_{p_i,p'_i}$$

By setting $r(p) - \sum_{i \in [n]} \sum_{p'_i \in \mathcal{S}_i} \lambda^i_{p_i,p'_i} (U_i(p_i, p_{-i}) - U_i(p'_i, p_{-i})) - \mu - k_p = 0$ we get that

$$\mu + \sum_{i \in [n]} \sum_{p'_i \in \mathcal{P}_i} \lambda^i_{p_i,p'_i} (U_i(p_i, p_{-i}) - U_i(p'_i, p_{-i})) \leq r(p)$$

since $k_s \geq 0$.

$\square$

## C  Omitted Proof of Section 3.1

**Lemma 5.** *Let a valid pricing profile* $p \in \mathcal{V}$. *Then* $b_p = 1$.

*Proof.* Since $p \in \mathcal{V}$ we know that $r(p) = 1$. Since $p$ is a valid pricing profile we know that $p_i = p^\star$ for each seller $i \in S(p^\star)$ with $s_i \leq p^\star$. As a result,

$$\sum_{p'_i \in \mathcal{S}_i} \hat{\lambda}^i_{p_i p'_i} \cdot (U_i(p_i, p_{-i}) - U_i(p'_i, p_{-i})) \quad = \quad \hat{\lambda}^i_{p_i p^\star} \cdot (U_i(p_i, p_{-i}) - U_i(p^\star, p_{-i})) \quad (\hat{\lambda}^i_{p_i p'_i} = 0 \text{ for } p'_i \neq p^\star)$$

$$= \quad 0 \quad \text{since } p_i = p^\star$$

For all the sellers $i$ with $s_i > p^\star$, by Definition 9 we have that $\sum_{p'_i \in \mathcal{S}_i} \hat{\lambda}^i_{p_i p'_i} = 0$ since $\hat{\lambda}^i_{p_i p'_i} = 0$ for all prices $p_i$ and $p'_i$.

$\square$

**Lemma 6.** *Let a non valid pricing profile* $p \notin \mathcal{V}$. *Then* $b_p \geq 1$.

*Proof.* Since the pricing profile $p \notin \mathcal{V}$ is not valid, by Definition 7 we get that $r(p) = 0$. As a result, we need to establish that $\sum_{i \in [n]} \sum_{p'_i \in \mathcal{S}_i} \hat{\lambda}^i_{p_i p'_i} \cdot (U_i(p'_i, p_{-i}) - U_i(p_i, p_{-i})) \geq 1$. To simplify notation let $S(p^\star)$ denote the set of sellers with marginal cost less than $p^\star$, $S(p^\star) = \{i \in [n] \ : \ s_i \leq p^\star\}$. Notice that by Definition 9, $\hat{\lambda}^i_{s_i s'_i} = 0$ for all $i \notin S(p^\star)$. To simplify notation we denote $m := |S(p^\star)|$. Let $i^\star \in [n]$ be the seller $i \in S(p^\star)$ with the highest price $p_i$, $i^\star := \mathrm{argmax}_{i \in S(p^\star)} p_i$. In case there are multiple sellers with price $p_{i^\star}$, we consider $i^\star$ to be the one with the lowest index.

Up next we show that for any non-valid pricing profile $p \notin \mathcal{V}$,

$$\sum_{i \in [n]} \sum_{p'_i \in \mathcal{S}_i} \hat{\lambda}^i_{p_i p'_i} \cdot (U_i(p'_i, p_{-i}) - U_i(p_i, p_{-i})) \geq 1.$$

We establish the latter claim for the following mutually exclusive cases:

1. $p_{i^\star} \leq p^\star$

2. $p_{i^\star} \geq p^\star + 1$.

Let us start with the case $p_{i^\star} \leq p^\star$. Since $p \notin \mathcal{V}$ we know that there exists an seller $i \in [n]$ such that $p_i < p^\star$ since otherwise $p \in \mathcal{V}$. Notice that any seller $i \in \mathcal{S}(p^\star)$ that set $p^\star$ as its price, is ensured to sell its good. This is because all produce $i \notin \mathcal{S}(p^\star)$ must essentially set a price $p_i > p^\star$ while there are exactly $m$ buyer willing to pay price $p^\star$ for the good. As a result, for any seller $i \in [n]$ we know that

$$\sum_{p_i' \in \mathcal{S}_i} \hat{\lambda}^i_{p_i p_i'} \cdot (U_i(p_i', p_{-i}) - U_i(p_i, p_{-i})) \quad = \quad \hat{\lambda}^i_{p_i p^\star} \cdot (U_i(p_i', p_{-i}) - U_i(p^\star, p_{-i}))$$

$$\geq \quad ((p^\star - s_i) - (p_i - s_i)) = p^\star - p_i$$

Since there exists at least one seller $i \in [n]$ with $p_i < p^\star$ and $\hat{\lambda}^i_{s_i s_i'} = 0$ for all $i \notin [n]$ we are ensured that

$$\sum_{i \in [n]} \sum_{p_i' \in \mathcal{S}_i} \hat{\lambda}^i_{p_i p_i'} \cdot (U_i(p_i', p_{-i}) - U_i(p_i, p_{-i})) \geq 1.$$

Let us now proceed with the case where $p_{i^\star} \geq p^\star + 1$. We first argue that $U_{i^\star}(p_{i^\star}, p_{-i^\star}) = 0$-the latter is formally established in Claim 1

**Claim 1.** *Let the seller* $i^\star = \mathrm{argmax}_{i \in [n]} p_i$ *(in case of ties* $i^\star \in [n]$ *is the one with lowest index). Then* $U_{i^\star}(p_{i^\star}, p_{-i^\star}) = 0$.

*Proof.* Since $p^\star$ is the maximum market clearing price then $p^\star + 1$ is not a market clearing price and thus $p^\star = b_j$ for some buyer $j \in [n]$ or $p^\star + 1 = s_i$ for some provider $i \in [n]$. Since all $s_i$ and $b_j$ lie in different positions the latter two cases are also mutually exclusive.

Let us start with the case $p^\star = b_j$ for some buyer $j \in [n]$. Since $p_{i^\star} \geq p^\star + 1$ we are ensured that there are at most $m - 1$ buyers willing to buy the good at price $p_{i^\star} \in [n]$. However since $i^\star \in \mathrm{argmax}! p_i$ and at the same time admits the lowest index, will not sell its good (notice that enter the market in decreasing order with respect to their $b_j$ and break ties among sellers lexicographically). Thus $U_{i^\star}(p_{i^\star}, p_{-i^\star}) = 0$. $\square$

Up next we consider the mutually exclusive cases $p^\star + 1 = b_j$ and $p^\star + 1 = s_i$ and separately establish that

$$\sum_{i \in [n]} \sum_{p_i' \in \mathcal{S}_i} \hat{\lambda}^i_{p_i p_i'} \cdot (U_i(p_i', p_{-i}) - U_i(p_i, p_{-i})) \geq 1.$$

We first start with the case $p^\star + 1 = b_j$ for some buyer $j \in [n]$. Let $i^\star \in [n]$ be the seller $i \in S(p^\star)$ with the highest price $p_i$, $i^\star := \mathrm{argmax}_{i \in S(p^\star)} p_i$. In case there are multiple sellers with price $p_{i^\star} \in [k]$ the $i^\star$ is the one with the lowest index among them.

Let us assume that $p_{i^\star} > p^\star$, byClaim 1 we are ensured that $U_{i^\star}(p_{i^\star}, p_{-i^\star}) = 0$. In case seller $i^\star \in [n]$ had selected price $p^\star \in [n]$, it would have sold its good since at price $p^\star \in [n]$ there are $m$ buyers and $m$ sellers willing to sell. Thus, $U_{i^\star}(p^\star, p_{-i^\star}) = p^\star - s_{i^\star}$. As a result we get that

$$\hat{\lambda}^i_{p_{i^\star} p^\star} \cdot (U_{i^\star}(p^\star, p_{-i^\star}) - U_i^\star(p_{i^\star}, p_{-i^\star})) = (2nk)^{2np_{i^\star} - i^\star} \cdot (p^\star - s_{i^\star}) \geq (2nk)^{2np_{i^\star} - i^\star}$$

Now let a seller $i \in [n]$ such that $p_i =_{i^\star}$. By definition of $i^\star$ we get that $i \geq i^\star + 1$. Thus,

$$\hat{\lambda}^i_{p_i p^\star} \cdot (U_i(p^\star, p_{-i}) - U_i(p_i, p_{-i})) \geq (2nk)^{2np_{i^\star} - i^\star - 1} \cdot (p^\star - p_{i^\star}) \geq -(2nk)^{2np_{i^\star} - i^\star - 1} k$$

Now let a seller $i \in [n]$ such that $p_i \leq p_{i^\star} - 1$. In this case

$$\hat{\lambda}^i_{p_i p^\star} \cdot (U_i(p^\star, p_{-i}) - U_i(p_i, p_{-i})) \geq (2nk)^{2n(p_{i^\star} - 1) - 1} \cdot (p^\star - p_{i^\star} + 1) \geq -(2nk)^{2np_{i^\star} - 2n - 1} k$$

As a result, we overall get that

$$\hat{b}_p \quad := \quad r(p) - \sum_{i \in S(p^\star)} \sum_{p_i' \in \mathcal{S}_i} \hat{\lambda}^i_{p_i p_i'} \cdot (U_i(p_i, p_{-i}) - U_i(p_i', p_{-i}))$$

$$= \quad (2nk)^{2np_{i^\star} - i^\star} - n(2nk)^{2np_{i^\star} - i^\star - 1} k - n(2nk)^{2np_{i^\star} - 2n - 1} k$$

$$= \quad (2nk)^{2np_{i^\star} - i^\star} \left(1 - \frac{1}{2} - nk(2nk)^{i^\star - 2n - 1}\right) \geq 1$$

We now consider the case where $p^\star + 1 = s_i$ for some seller $i \in [n]$. To simplify notation we denote this seller as Next. Following the exact same steps as in the previous case, we can prove that

$$\sum_{i \neq i^\star} \sum_{p_i' \in \mathcal{S}_i} \hat{\lambda}^i_{p_i p_i'} \cdot \left( U_i(p_i, p_{-i}) - U_i(p_i', p_{-i}) \right) \geq -n(2nk)^{2np_{i^\star} - i^\star - 1} k - n(2nk)^{2np_{i^\star} - 2n - 1} k$$

Let us now consider $\lambda^{\text{next}}_{p_i(p^\star+1)} = (2nk)^{2np_i + \text{next}} / \epsilon$ where $\epsilon > 0$ is the small positive constant discussed in Section 2. Up next we establish Lemma 6 for the following mutually exclusive cases:

- $\underline{p_{i_\star} = p^\star + 1 \text{ and } p_{\text{next}} = p^\star + 1}$

  In this case $U_{i^\star}(p) = 0$ since there are $m$ sellers before seller $i^\star \in [n]$ with higher priority (next has higher index than $i^\star$). At the same time $U_{i^\star}(p^\star, p_{-i_\star}) = p^\star - s_{i^\star}$. Finally in case next selects $p^\star + 1$ as its price it gets the good and thus $U_{next}(p^\star + 1, p_{-\text{next}}) = p^\star + 1 - s_{\text{next}} = \epsilon$. Combining all the above we get that

  $$\sum_{i \in \{i^\star, \text{next}\}} \sum_{s_i' \in \mathcal{S}_i} \hat{\lambda}^i_{p_i p_i'} \left( U_i(p_i, p_{-i}) - U_i(p_i', p_{-i}) \right) \geq (2nk)^{2np^\star - i^\star}$$

- $\underline{p_{i_\star} = p^\star + 1 \text{ and } p_{next} \geq p^\star + 2}$

  In this case $U_{i^\star}(p) = p^\star + 1 - s_{i^\star}$ since $i^\star \in [n]$ sells its good due to the fact that there are at least $m$ buyers willing to buy the good at price $p^\star + 1$. Similarly $U_{i^\star}(p^\star, p_{-i_\star}) = p^\star - s_{i^\star}$. Also notice that next does not sell its good if it sets with price $p_{\text{next}}$ but sell its if it sets price $p^\star + 1$. As a result, $U_{\text{next}}(p_{\text{next}}, p_{-i}) = 0$ and $U_{\text{next}}(p^\star + 1, p_{-i}) = \epsilon$. Combing all the above we get

  $$\begin{aligned}
  \sum_{i \in \{i^\star, \text{next}\}} \sum_{s_i' \in \mathcal{S}_i} \hat{\lambda}^i_{p_i p_i'} \left( U_i(p_i', p_{-i}) - U_i(p_i, p_{-i}) \right) &\geq -(2nk)^{2np^\star + 2n - i^\star} + (2nk)^{2np_{\text{next}} - \text{next}} \\
  &\geq -(2nk)^{2np^\star + 2n - i^\star} + (2nk)^{2np^\star + 4n - \text{next}}
  \end{aligned}$$

- $\underline{p_{i_\star} \geq p^\star + 2 \text{ and } p_{next} \leq p_{i_\star}}$

  In this case notice that $U_{i^\star}(p) = 0$ since there are $m$ sellers with higher priority that seller $i \in [n]$. At the same time $U_{i^\star}(p^\star, p_{-i_\star}) = p^\star - s_{i^\star}$ since seller next will never report price $p^\star$ ($s_{\text{next}} > p^\star$) thus seller $i^\star$ always sells its good at price $p^\star \in [n]$. Similarly as before we get that $U_{\text{next}}(p^\star + 1, p_{-\text{next}}) = \epsilon$ and $U_{\text{next}}(p_{\text{next}}, p_{-\text{next}}) \leq p_{\text{next}} - s_{\text{next}} = p_{\text{next}} - (p^\star + 1 - \epsilon)$. Combining all the above we get that

  $$\begin{aligned}
  \sum_{i \in \{i^\star, \text{next}\}} \sum_{s_i' \in \mathcal{S}_i} \hat{\lambda}^i_{p_i p_i'} \left( U_i(p_i', p_{-i}) - U_i(p_i, p_{-i}) \right) &\geq (2nk)^{2np_{\text{next}} - i^\star} - \frac{k}{\epsilon} (2nk)^{2np_{\text{next}} - \text{next}} \\
  &= (2nk)^{2np_{\text{next}} - i^\star} \left( 1 - \frac{k}{\epsilon} (2nk)^{-1} \right) \\
  &= (2nk)^{2np_{\text{next}} - i^\star} \left( 1 - \frac{1}{2\epsilon n} \right) \\
  &= \frac{1}{2} (2nk)^{2np_{\text{next}} - i^\star} \text{ for } \epsilon = 1/n
  \end{aligned}$$

- $\underline{p_{i_\star} \geq p^\star + 2 \text{ and } p_{\text{next}} > p_{i_\star}}$

  In this case $U_{i^\star}(p) \leq p_{i^\star} - s_{i^\star}$ and $U_{i^\star}(p^\star, p_{-i_\star}) = p^\star - s_{i^\star}$. At the same time notice that $U_{\text{next}}(p^\star + 1, p_{-\text{next}}) = \epsilon$ since seller next is able to sell its good at price $p^\star + 1$. While seller next can never sell its good at price $p_{\text{next}} \in [k]$ since there $m$ other sellers with higher priority. Thus, $U_{\text{next}}(p_{\text{next}}, p_{-\text{next}}) = 0$. Combining all the above we get

$$\sum_{i\in\{i^\star,\text{next}\}}\sum_{s_i'\in\mathcal{S}_i}\hat{\lambda}^i_{p_ip_i'}\left(U_i(p_i',p_{-i})-U_i(p_i,p_{-i})\right) \geq -k(2nk)^{2n(p_\text{next}-1)-i^\star}+\frac{1}{\epsilon}(2nk)^{2np_\text{next}-\text{next}}\cdot\epsilon$$

$$= (2nk)^{2np_\text{next}}\left(-k(2nk)^{-2n-i^\star}+(2nk)^{-\text{next}}\right)$$

$$= \frac{1}{2}(2nk)^{2np_\text{next}-\text{next}}$$

As a result, we overall get that

$$\sum_{i\in\{i^\star,\text{next}\}}\sum_{s_i'\in\mathcal{S}_i}\hat{\lambda}^i_{p_ip_i'}\left(U_i(p_i',p_{-i})-U_i(p_i,p_{-i})\right)\geq\frac{1}{2}(2nk)^{2np_{i^\star}-i^\star-1}$$

At the same time using the exact same arguments as in the case $p^\star = b_j$ for some buyer $j\in[n]$, we establish that

$$\sum_{i\neq i^\star,\text{next}}\sum_{p_i'\in\mathcal{S}_i}\hat{\lambda}^i_{p_ip_i'}\cdot\left(U_i(p_i,p_{-i})-U_i(p_i',p_{-i})\right)\geq -n(2nk)^{2np_{i^\star}-i^\star-3}k-n(2nk)^{2np_{i^\star}-2n-1}k$$

Putting everything together we get that

$$\sum_{i\in[n]}\sum_{p_i'\in\mathcal{S}_i}\hat{\lambda}^i_{p_ip_i'}\cdot\left(U_i(p_i,p_{-i})-U_i(p_i',p_{-i})\right)\geq\frac{1}{2}(2nk)^{2np_{i^\star}-i^\star-1}-n(2nk)^{2np_{i^\star}-i^\star-3}k-n(2nk)^{2np_{i^\star}-2n-1}k\geq 1$$

$$\square$$

## D Additional Experimental Evaluations

All experiments were conducted in Apple M4 Pro and the Hedge algorithm was run with step-size $\gamma = 0.1$.

We also evaluate the Hedge algorithm in the following set-up. We consider the demand curve $\mathcal{D}(p) = -20p + 100$ for $p \in [1, 5]$ with a $0.2$ discretization. We consider $4$ different supply curves parametrized by $m \in \{0, 100\}$,

$$S_m(p) = \begin{cases} m & \text{if } p \leq 4.8 \\ \frac{100 - m}{0.2} \cdot p + \frac{5m - 480}{0.2} & \text{otherwise} \end{cases}$$

In Figure 6 we consider $m \in \{10, 30, 60, 80\}$. Each different curve $\mathcal{S}_m$ admits a different intersection point with the demand curve (see Figure 6). In Figure 7 we see that if all sellers use a no-swap regret algorithm the average selling price converges to the respective market clearing price. The average selling price at each round $t$, is the average price of the realized trades.

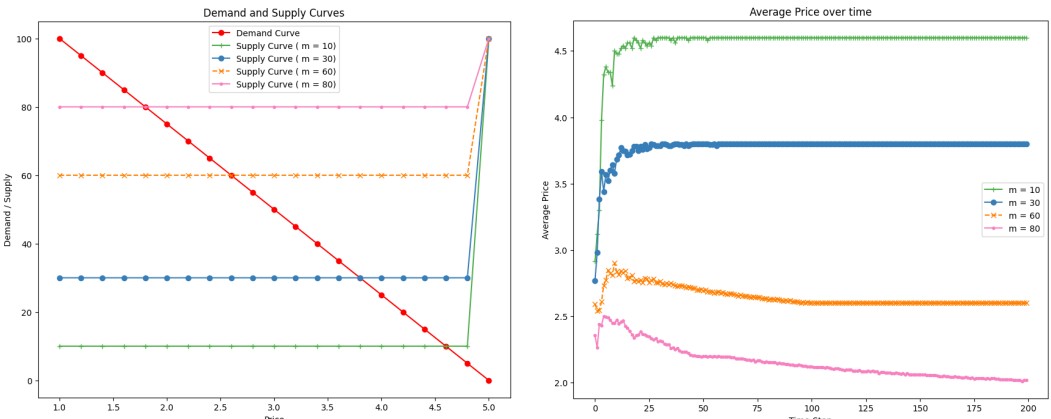

Figure 6: Supply and Demand Curves        Figure 7: Average Selling Price

In all of our experimental evaluation we used step-size $\gamma = 0.1$ for the Hedge algorithm.

### D.1 No-Swap Regret Dynamics

We perform the exact same experimental evaluation for the no-swap regret algorithm of Blum et al. [7]. In our implementation we used the Hedge algorithm with step-size $\gamma = 0.1$ as our base no-regret algorithm. In all the above experimental evaluation the resulting no-swap regret dynamics converge to the market clearing price, something that is to be expected due to Theorem 1.

As in Section 4, we first consider the case $n = 2$ sellers and buyers where $(s_1, s_2) = (0, 0)$ and $(b_1, b_2) = (5, \lambda)$. In this instance the highest market clearing price is $p^\star = \lambda$. Figures 8, 9 and 10 verify that the resulting no-swap regret dynamics converge to the respective market clearing price of each case.

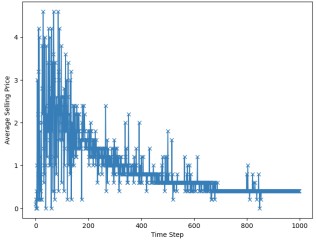 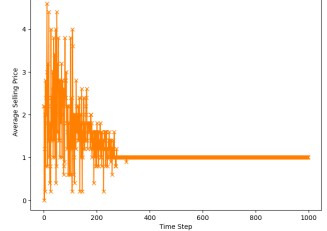 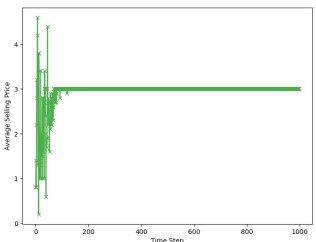

Figure 8: $\lambda = 0.2$        Figure 9: $\lambda = 1$        Figure 10: $\lambda = 3$

Next we consider the case $n = 100$ sellers and sellers with demand curve $\mathcal{D}(p) = -20p + 100$ for $p \in [0, 5]$ and supply curves, $\mathcal{S}_{\text{linear}}(p) := p/5$, $\mathcal{S}_{\text{quad}}(p) := p^2/0.25$ and $\mathcal{S}_{\text{linear}}(p) := p/5$, $\mathcal{S}_{\text{sqrt}}(p) := 100\sqrt{p/5}$. Each supply curve intersects with the demand curve at a different price, resulting in different market clearing prices (see Figure 12). Figure 11 verifies that the resulting no-swap regret dynamics converge to the respective market clearing price of each case.

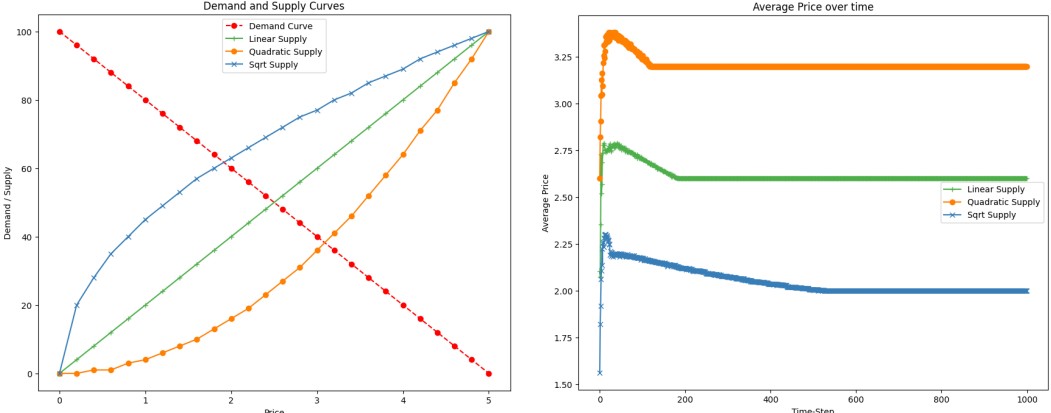

Figure 11: Supply and Demand Curves     Figure 12: Average Selling Price

Finally we consider the demand curve $\mathcal{D}(p) = -20p + 100$ for $p \in [1, 5]$ with a $0.2$ discretization. We consider $4$ different supply curves parametrized by $m \in \{10, 30, 60, 80\}$,

$$S_m(p) = \begin{cases} m & \text{if } p \le 4.8 \\ \frac{100-m}{0.2} \cdot p + \frac{5m-480}{0.2} & \text{otherwise} \end{cases}$$

In Figure 14 we see that if all sellers use the no-swap regret algorithm of Blum et al. [7], the average price converges to the respective market clearing price.

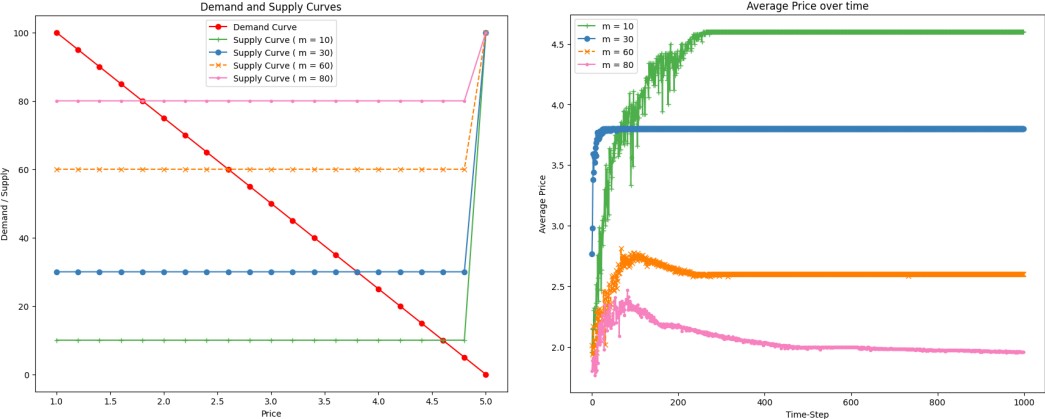

Figure 13: Supply and Demand Curves     Figure 14: Average Selling Price

