# OpenReview forum: "Explaining the Law of Supply and Demand via Online Learning"
_NeurIPS.cc/2025/Conference — NeurIPS 2025 poster_

### Official Review · Reviewer_7ou3 · 2025-07-02

**Clarity:** 2
**Significance:** 2
**Originality:** 2
**Rating:** 3
**Confidence:** 3

**Summary:**

This paper studies the Law of Supply and Demand from a novel perspective, framing it as an outcome of adaptive learning processes among sellers in a perfectly competitive market. The core contribution is that the authors demonstrate how, if each seller employs a no-swap regret algorithm to set their individual selling price, the collective pricing dynamics converge to the market-clearing price. Experimental results are provided to demonstrate the effect.

**Questions:**

Could you elaborate on what distinguishes these "specific classes", in the statement of "the negative result of Theorem 3 may be circumvented by specific classes of no-regret algorithms"?

**Ethical Concerns:**

["NO or VERY MINOR ethics concerns only"]

**Final Justification:**

The main positive result Theorem 1 and negative result Theorem 2 are indeed interesting. My concern is Theorem 2's **contradiction** with the experimental results, which is explained only with the vague term “likely” in the discussion. Just as they provided experimental evidence for Theorem 1, I would expect a similar experimental demonstration for Theorem 2’s non-convergence result, especially since they explicitly state in Line 293: “We first consider the family of instances of the pricing game constructed to **establish** Theorem 2.” Yet the figure’s result directly conflicts with Theorem 2, which is not a tangential point in my opinion, and “likely” is not an acceptable explanation to me for this discrepancy.

The convergence path and stability during the learning process could still exhibit variance. Including error bars or confidence intervals, especially for the initial phases of learning, would provide a more complete picture of the algorithms' performance.

Finally, including experiments on other common no-regret algorithms beyond Hedge would make the work more convincing. Demonstrating results on just one algorithm is not persuasive enough.

That said, I’m willing to raise my rating from 2 to 3, but I will not champion the acceptance of this paper.

**Limitations:**

yes

**Quality:**

2

**Strengths And Weaknesses:**

On the positive side, this paper offers a fresh view on a fundamental economic principle by connecting it to the field of online learning and game theory. I found the problem presented by the paper interesting and relevant. In general, this paper proposed an innovative research topic.

On the other hand, the paper notes that while general no-regret algorithms don't theoretically guarantee convergence to the market-clearing price, the experimental results for Hedge show convergence, suggesting it might be circumvented by "specific classes of no-regret algorithms." This argument is not convincing. Could you elaborate on what distinguishes these "specific classes"? For example, including experimental evaluations of other common no-regret algorithms beyond Hedge to further explore this phenomenon. It would be beneficial to see how other common no-regret or online learning algorithms perform, even if they are not expected to converge to CE, and to elaborate on what constitutes these "specific classes."

In addition, the paper states that "All of our evaluated online learning dynamics converge to the market clearing price meaning that there is no variance on the final outcome." While this might be true for the final outcome, the convergence path and stability during the learning process could still exhibit variance. Including error bars or confidence intervals, especially for the initial phases of learning, would provide a more complete picture of the algorithms' performance.

Another problem with the paper is its presentation. Unfortunately, the paper doesn’t flow very well. The writing of the paper seems to be a bit rushed. There are typos, and grammatical and formatting errors. For example, in the abstract, “the number of sellers willing to …… equals the number of sellers (the authors meant buyers?) ……”

---

> ### Author Rebuttal · Authors · 2025-07-27
>
> Thank you very much for all your work and for appreciating the innovative aspects of our work. Up next we address your main concerns.
>
>
> **On the other hand, the paper...to elaborate on what constitutes these "specific classes."**
>
>
> With specific classes we refer to the *mean-based algorithms*. Notice that we write *"Ιt is likely that a certain classes of no-regret algorithms, *such as mean-based algorithms*, may be able to always converge to the market clearing price"*.
>
> Mean-based algorithms [4] is a natural class of no-regret algorithms (including Hedge) with the following property: if the cumulative reward of two actions differs by at most $\mathcal{O}(\gamma \cdot T)$ then the respective probabilities differ by at most by $\mathcal{O}(\gamma)$. Mean-based nο-regret algorithms are known to avoid degenerate CCE in various games (e.g. auctions, potential games [1,2,3]). As a result, we conjecture that this may be the case for Supply and Demand games. The latter is aligned with our experimental evaluations.
>
> In the revised version of our work, we will add the above discussion to clarify the situation. We will also add additional experimental evaluations of no-regret algorithms (FTRL, Online Mirror Descent, Regret Matching etc.). We will additionally correct the typos and add error bars.
>
> **Remark:** The main result of our work is formally establishing that *Correlated Equilibrium* (CE) corresponds to the market clearing price in the case of Supply and Demand games. We remark that the discussion on no-regret algorithms does not constitute neither an argument nor a result of the current paper, but rather a conjecture and a suggestion for future research direction.
>
>
> We hope the latter clarify your concerns and will help reconsider your score.
>
>
> [1] Deng et al. *Nash convergence of mean-based learning algorithms in first price auctions*.
>
> [2] Feng et al. *Convergence Analysis of No-Regret Bidding Algorithms in Repeated Auctions*.
>
> [3]  Heliou et al. *Learning with Bandit Feedback in Potential Games*.
>
> [4] Braverman et al. *Selling to a No-Regret Buyer*.

---

> > ### Comment · Reviewer_7ou3 · 2025-08-01
> >
> > Thank you for the clarification. However, I remain unconvinced by the use of the term "likely" in the statement. Furthermore, the discussion on no-regret algorithms is presented as a formal result—specifically, Theorem 2—which appears to be on equal status with the main result, Theorem 1. This contradicts your statement that "the discussion on no-regret algorithms does not constitute either an argument or a result of the current paper."
> >
> > Additionally, there are further concerns regarding the experimental evaluations. Given these points, I maintain my current assessment: the paper requires further revision and, in its current form, is not ready for publication.

---

> > > ### Author Response · Authors · 2025-08-02
> > > **Author Response**
> > >
> > > Thank you very much for your response. However we believe that there is still a misunderstanding. In particular,
> > >
> > > 1. Theorem 2 is a formal **negative** result of the paper establishing that there are CCEs not corresponding to the market clearing price.
> > >
> > > 2. Our conjecture/discussion is on the **positive** side -  specific classes of no-regret algorithms (e.g. mean-based algorithms) converge to the market clearing price.
> > >
> > > The latter statements are not contradictory since specific classes of no-regret algorithms can avoid bad CCEs. However we would be willing to remove the respective statement in the conclusion if you find it confusing.
> > >
> > > The experimental evalautions are only side results. The main result of the paper is formally establishing that no-swap algorithms converge to the market clearing price.
> > >
> > > Could you share your thoughts on the main results of the paper?
> > >
> > > Please let us whether the discussion above helps you reconsider your score or you need futher clarifications.

---

> > > > ### Comment · Reviewer_7ou3 · 2025-08-08
> > > >
> > > > The results on no-swap regret algorithms look good to me, thank you for your emphasis, I will take them into account during the AC-Reviewer discussion phase.

---

### Official Review · Reviewer_6iRp · 2025-07-03

**Clarity:** 3
**Significance:** 2
**Originality:** 3
**Rating:** 3
**Confidence:** 4

**Summary:**

The paper proves that in a repeated pricing game, if all sellers use no-swap regret algorithms, the resulting play converges to correlated equilibria that place all probability on profiles where prices equal the market-clearing price. The proof is an interesting application of dual-fitting. The paper also shows that the assumption that the players use no-swap regret algorithms is necessary, since with regular no-regret algorithms, the game converges to a coarse correlated equilibrium, and there are cases where such equilibria does not correspond to a market clearing price. However, such cases are not observed in numerical experiments.

**Questions:**

How robust are the results to noise, incomplete information, or heterogeneous learning rates?

**Ethical Concerns:**

["NO or VERY MINOR ethics concerns only"]

**Limitations:**

yes

**Quality:**

3

**Strengths And Weaknesses:**

Strengths:
* A game dynamics foundation for the law of supply and demand.
* To the best of my knowledge, the use of dual-fitting to characterize the equilibria is novel.

Weaknesses:
* While theoretical results are interesting and non-trivial, they might not be at the level of a NeurIPS paper.
* No-swap regret might be unrealistic in practice.
* Assumes full information, also somewhat unrealistic.
* Model is a bit too simplistic.

---

> ### Author Rebuttal · Authors · 2025-07-27
>
> Thank you very much for all your work and for appreciating the motivation of our work. Up next we adrress your main concerns.
>
> **1.While the theoretical results are interesting and non-trivial, they might not meet the standard of a NeurIPS paper.**
>
> The Law of Supply and Demand is arguably the most fundamental principle in modern economics. We believe that establishing that it can emerge from a learning process among competing sellers is both a novel and significant contribution. We remark that the latter is not guaranteed—since prices could perpetually oscillate. Our results show that, under mild assumptions, convergence to market equilibrium occurs, which we see as a meaningful insight into market dynamics.
>
> **2.No-swap regret might be unrealistic in practice.**
>
> No-swap regret algorithms have been extensively studied in the literature as models for the learning behavior of competing agents [1,2]. More recently, several works have shown important advantages of no-swap regret, especially in terms of robustness to exploitation [3,4,5]. The latter suggest that no-swap regret is a compelling modeling choice in strategic multi-agent settings.
>
> References:
>
> [1] Hart & Mas-Colell, *A Simple Adaptive Procedure Leading to Correlated Equilibrium*
>
> [2] Foster & Vohra, *Calibrated Learning and Correlated Equilibrium*
>
> [3] Arunachaleswaran et al., *Swap Regret and Correlated Equilibria Beyond Normal-Form Games*
>
> [4] Arunachaleswaran et al., *Pareto-Optimal Algorithms for Learning in Games*
>
> [5] Mansour et al., *Strategizing against Learners in Bayesian Games*
>
> **3. Assumes full information, also somewhat unrealistic.**
>
> Our results do not require any *full information* assumption. If agents use any no-swap learning algorithm with *bandit feedback* (only the revenue of the selected price is learned), they will converge to a CE and our results will apply. The same holds in case of noisy perturbations on the reward.
>
> **4.Model is a bit too simplistic.** Since our model captures the essence of Law of Supply and Demand, we believe that simplicity is a plus.
>
>
> **Questions**
>
> **How robust are the results to noise, incomplete information, or heterogeneous learning rates?**
>
> Our results continue to apply in case of noise / incomplete information / heterogeneous learning rates and even heterogenous algorithms. Our main result establishes any CE is a market clearing price. We remark that convergence to CE is achieved even if agents use different no-swap regret algorithms. At the same time, there are various no-swap regret algorithms that work with noisy perturbations of the reward or even with *bandit feedback* (only the revenue of the selected price is revealed to the seller). Even in such limited information settings, if agents use such no-swap regret algorithms they will converge to CE and thus our results apply.
>
>
>
> We hope the latter clarify your concerns and encourage a reconsideration of your score. We would be happy to provide further clarifications if needed.

---

### Official Review · Reviewer_XyhP · 2025-07-05

**Clarity:** 4
**Significance:** 4
**Originality:** 4
**Rating:** 5
**Confidence:** 4

**Summary:**

This paper revisits the law of supply and demand through the lens of game theory and online learning. The authors model a pricing game in which sellers independently set prices to maximize their individual revenues. They prove that if sellers adopt no-swap regret algorithms, the resulting price dynamics converge to the market-clearing price. In contrast, they construct a counterexample showing that coarse correlated equilibria (CCE), which arise under general no-regret dynamics, may fail to align with the market-clearing price. Empirical results demonstrate that both no-regret and no-swap regret algorithms typically lead to convergence in practice.

**Questions:**

1) The analysis assumes a discrete set of prices. Could the framework be extended to continuous prices? How sensitive are the theoretical guarantees to the choice of discretization, and would convergence behavior change in the continuous case?
2) How would the convergence results be affected by noise in seller feedback or slight uncertainty in buyer valuations?
3) Theorem 2 presents a negative result where no-regret dynamics may fail to reach the market-clearing price. It would be insightful to include experimental scenarios where this failure is observed, highlighting the contrast with the success of no-swap regret algorithms.

**Ethical Concerns:**

["NO or VERY MINOR ethics concerns only"]

**Final Justification:**

I lean towards acceptance primarily because of the novel perspective it offers on understanding the law of supply. While there are many potential extensions, the paper makes an interesting contribution.

**Limitations:**

Yes.

**Quality:**

3

**Strengths And Weaknesses:**

Strengths
1) The paper provides a novel theoretical foundation for a classical economic principle by framing it through the lens of modern online learning dynamics.
2) The proposed pricing game, in which sellers act strategically while buyers follow deterministic purchasing rules, is well-motivated and effectively captures essential characteristics of perfectly competitive markets.
3) A key technical contribution is the use of a dual-fitting technique to show that all correlated equilibria correspond to valid pricing profiles, which is a novel approach and possibly of broader interest.
4) Simulation results support the theoretical claims demonstrating convergence to the market-clearing price across a variety of market configurations.

Weaknesses
1) Section 1.1 reviews relevant literature by summarizing key ideas, but lacks a systematic comparison between this work and prior studies.
2) The model assumes that only sellers are strategic, while buyers arrive sequentially and always purchase from the lowest-priced seller. This assumption simplifies more realistic buyer behaviors, such as strategic decision-making and preference heterogeneity.
3) The experimental evaluation is mainly illustrative and does not assess performance under large-scale or noisy environments. Moreover, it focuses only on two algorithms without broader comparisons across other online algorithms.

---

> ### Author Rebuttal · Authors · 2025-07-27
>
> Thank you very much for all your work and the positive feedback on our work. Up next we address your main concerns.
>
>
>
> **1.Section 1.1 reviews...this work and prior studies.**
>
> In the revised version of our work, we will discuss in further detail the related literature.
>
> **2.The model assumes that...preference heterogeneity.**
>
> The law of Supply and Demand assumes that all provided goods are *indistiguishable*. When buyer preferences vary (preference heterogeneity) the law of Supply and Demand no longer holds. Also notice that in case of indistinguishable goods, strategic buyer behavior reduces to simply purchasing from the cheapest available seller.
>
> **3.The experimental evaluation...online algorithms.**
> The primary focus of our work is to formally establish the law of supply and demand as an emergent behavior of learning dynamics among sellers. That being said, in the revised version, we will include additional experimental evaluations using a broader set of no-regret algorithms, including FTRL, Regret Matching, Online Mirror Descent, and others.
>
>
> Questions
>
> 1.**The analysis assumes...in the continuous case?**
>
> We believe that our results and techniques extend to the continuous price model by considering the respective LPs with infinite number of constraints. In the current work we chose the discrete price model since it leads to simpler analysis and better reflects practice where prices are typically discrete.
>
> 2.**How would the convergence results be affected by noise in seller feedback or slight uncertainty in buyer valuations?**
>
> The are not affected at all. There are no-swap regret algorithms that work under noisy perturbations on the payoffs or even bandit feedback (only the revenue of the selected price is observed). If agents adopt such algorithms the overall system will converge to a CE and thus our results apply.
>
> **3.Theorem 2 presents a negative result...no-swap regret algorithms.**
>
> In the revised version, we will present further experimental evaluations of various no-regret algorithms. However we conjecture that natural no-regret algorithms avoid bad Coarse Correlated Equilibria (CCE) and instead converge to the market-clearing price. Our negative result in Theorem 2 highlights the limitations of reasoning about general CCEs, suggesting that future analysis should focus on specific no-regret algorithms rather than on general CCEs.

---

> > ### Comment · Reviewer_XyhP · 2025-08-06
> >
> > Thank you for the response. I still have one question regarding strategic buyers and preference heterogeneity:
> >
> > The reply suggests that the law of supply and demand no longer holds under preference heterogeneity. I believe this is a mischaracterization. In fact, heterogeneous preferences are standard in many economic models and do not invalidate the law; rather, they lead to more complex, yet well-defined, aggregate demand behavior. Could the authors clarify whether the assumption of homogeneous, non-strategic buyers is made purely for tractability, and whether incorporating richer buyer behavior is considered out of scope for this work?
> >
> > Additionally, if the model can be extended to explicitly handle noise in seller feedback with formal theoretical analysis, it would further strengthen the paper’s contribution.

---

> > > ### Author Response · Authors · 2025-08-06
> > > **Response to Reviewer XyhP**
> > >
> > > Thank you for your response.
> > >
> > > **1.** We agree with the reviewer that there are extensions of the law of supply and demand for heterogenous goods. In this work we focus on the homogenous case as the natural first step. We agree that extending our results to the heterogenous case is a very interesting reseach direction and we will add it in the conclusion of the paper.
> > >
> > > **2.** Our results hold no matter noisy or partial feedback for the sellers. We establish that any CE is a market clearing price. Convergence to CE is achieved once each agent uses a no-swap regret algorithm (that may differ from agent to agent). There are various no-swap regret algorithms that operate under *bandit feedback* (each seller observes only the revenue of its selected price) or *noisy perturbations* of the reward. As a result, if all agents use such *limited information* no-swap regret algorithms, our results establish convergence to the market clearing price.
> > >
> > > We remark that our results persist even if some agents use limited-information and some other use full-information feedback no-swap regret algorithms.

---

### Official Review · Reviewer_CWGT · 2025-07-07

**Clarity:** 4
**Significance:** 3
**Originality:** 3
**Rating:** 5
**Confidence:** 3

**Summary:**

The paper studies convergence of prices in market settings, and specifically showing convergence to the so-called market clearing price, which is where demand meets supply. The game being played is fairly simple and requires sellers to set prices above their willingness to sell according to their strategy, and buyers to accept the price if it is above their willingness to buy. The first buyer gets the cheapest seller, the second the second-cheapest, and so on. Only sellers have strategies in this game, and they play a number of rounds. The aim is to minimise regret. They investigate the general classes of no-regret and no-swap regret algorithms. These are classes of online algorithms, which take past prices as input to produce the next price. They show that, when all sellers use a no-swap regret algorithm, then prices converge to the market clearing price which is a correlated equilibrium. However, they show by example that this does not always hold for coarse correlated equilibria, i.e. for more  no-regret algorithms. The latter is actually quite a suprising result. They then show that, in practice using a specific no-regret algorithm, convergence is still observed. In terms of contribution, existing works have shown converengence of these classes of algorithms, but they have not shown convergence to market clearing prices. At the same time, certain processes such as Walrasian tatonnement have been shown to converge to market clearing prices, but this is not generally shown for more general classes of online learning algorithms. So the contribution is to show the convergence to market clearing prices of a general class of algorithms.

**Questions:**

1. The sellers and buyers are ordered in terms of their valuation, and the paper states that this is "without loss of generality". However, in the definition of the game, i.e. Definition 3, buyers with lower index are able to choose "first", i.e. meaning that those buyers with the highest valuation are able to choose first and therefore get the lowest price. This reflects e.g. the double auctions protocol, where sellers with the lowest ask are matched with buyers with the highest bid. Surely, this choice is significant and would affect the results if, say, the buyer order was randomised? A brief note would be useful here.

2. According to Lemma 1, a market clearing price is (only?) guaranteed if marginal costs and prices "lies in different places". First, there is a grammatical error (should be plural i.e. "lie" not "lies"). Second, the expression is confusing and seems unusual (it might be a formal term I'm just not familiar with). Presumably it means they are all different. Maybe there is a better way (e.g. more formal way even) of defining this? More importantly, is this assumption required throughout, i.e. for all results in the paper? This was not clear. It seems somewhat restrictive.

(if space allows but less important):

3. Def 4 could possibly be standard, but I could not understand why the first term of the first equation is over expectation - i.e. p*_i is not a mixed strategy, and is also not used in the function U so why sample over the mixed strategy? Possibly to be consistent with the second equation?

**Ethical Concerns:**

["NO or VERY MINOR ethics concerns only"]

**Final Justification:**

Thanks to the autors for providing the response. I'm satisifed with the answers to my questions and have no further queries. I have also looked at the other reviews and none seem to identify any technical flaws with the paper nor issues around novelty. Hence, I stand by my initial score.

**Limitations:**

There are no obvious negative societal impacts from this work.

**Paper Formatting Concerns:**

No formatting issues found.

**Quality:**

4

**Strengths And Weaknesses:**

The paper is very well written, and largely clear. There is novelty and the contribution seem significant and relevant. However, I'm not an expert especially in the methods used for proving the results, and so was unable to check all the technical details. A few questions and minor issues:

- The sellers and buyers are ordered in terms of their valuation, and the paper states that this is "without loss of generality". However, in the definition of the game, i.e. Definition 3, buyers with lower index are able to choose "first", i.e. meaning that those buyers with the highest valuation are able to choose first and therefore get the lowest price. This reflects e.g. the double auctions protocol, where sellers with the lowest ask are matched with buyers with the highest bid. Surely, this choice is significant and would affect the results if, say, the buyer order was randomised? A brief note would be useful here.

- According to Lemma 1, a market clearing price is (only?) guaranteed if marginal costs and prices "lies in different places". First, there is a grammatical error (should be plural i.e. "lie" not "lies"). Second, the expression is confusing and seems unusual (it might be a formal term I'm just not familiar with). Presumably it means they are all different. Maybe there is a better way (e.g. more formal way even) of defining this? More importantly, is this assumption required throughout, i.e. for all results in the paper? This was not clear. It seems somewhat restrictive.

- Typo: Below Def 2: "sates" -> "states"

- L128: "an one-shot game" -> "a one-shot game" (I know it's confusing but see here for explanation: https://www.reddit.com/r/grammar/comments/8uctlm/is_it_a_one_or_an_one/)

- L156: "In a pricing games" -> "In a pricing game"

- Def 4 could possibly be standard, but I could not understand why the first term of the first equation is over expectation - i.e. p*_i is not a mixed strategy, and is also not used in the function U so why sample over the mixed strategy? Possibly to be consistent with the second equation?

- In Def 4, o(T) is not defined.

---

> ### Author Rebuttal · Authors · 2025-07-27
>
> Thank you very much for all the very detailed review and positive feedback on our paper. In the revised version of our paper, we will fix all the typos and minor comments.
>
> Up next we address your main questions and concerns.
>
> 1. **The sellers...A brief note would be useful here.** You are right and we apologize for the confusion. Wlog was supposed to refer on the ordering of the sellers. We will add a brief note to clarify.
>
> 2. **According to Lemma 1...somewhat restrictive.** We believe that our results hold even without this assumption however in such a case one would need to deal with tedious technical details. We chose to use this assumption that greatly simplifies the analysis since in case of sufficiently large discretization, it is natural to assume that each seller admits a slightly different price.
>
> 3. **Def 4 could possibly be standard...Possibly to be consistent with the second equation?** You are right that $p_i^\star$ is a derministic price however observe that all other agents sample their prices $p_{-i}^t$ according to the probablity distribution $x_{-i}^t$ and that is why we use the expectation.

---

### Decision · Program_Chairs · 2025-09-17

**Decision:**

Accept (poster)

**Comment:**

Two of the reviewers are largely positive about the paper, noting several strengths: new theoretical foundations behind a well-known economic principle through modern online learning; an interesting model of a pricing game; technical contributions that provide useful characterizations of the set of correlated equilibria of the game through a novel approach; and, results are complemented by simulations. Overall, the strengths of the paper are significant. The reviewers praise novelty, clarity, and significance.

Two of the reviews propose a weak reject. After reading the reviews, and discussion, and taking a quick look at the paper myself, I believe the two weak rejects do not provide sufficient grounds for the paper to be rejected. To elaborate a bit more:
- one reviewer states that "results (...) might not be at the level of a NeurIPS paper" and "model is a bit too simplitic", without further justfication. I cannot act on these comments, since they do not provide a reason for such evaluation and seem to disagree with the majority of the review team.
- the same reviewer mentions that "No-swap regret might be unrealistic in practice" and "assumes full information, also somewhat unrealistic.", but these assumptions are common in the literature and in theory papers, so I do not see this as an issue.
- lastly, the second negative reviewer only had negative comments that were mainly disconnected from the main directions of the paper. One complains about proposed future work, claiming that it was not formal enough/in theorem form, but this was not a contribution of the paper so this is not an issue. Another comment was with respect to the neurips checklist, asking for error bars for their convergence guarantees; since said convergence guarantees are in the limit and deterministic, error bars are not needed to evaluate the authors' contributions; what the reviewer is asking is a different contribution, and cannot be construed as a valid criticism of the authors' responses on the checklist. Finally, there seemed to be confusion about the technical results, with Theorem 2, which is a formally proven negative result, which is conflated with a mostly unrelated point in the review.

Based on this, I recommend the paper for acceptance.